# Implementation and Continuous Monitoring of an Electronic Health Record Embedded Readmissions Clinical Decision Support Tool

**DOI:** 10.3390/jpm10030103

**Published:** 2020-08-26

**Authors:** David Gallagher, Congwen Zhao, Amanda Brucker, Jennifer Massengill, Patricia Kramer, Eric G. Poon, Benjamin A. Goldstein

**Affiliations:** 1Hospital Medicine Programs, Division of General Internal Medicine, Duke University, DUMC 100800, Durham, NC 27710, USA; 2Department of Biostatistics and Bioinformatics, Duke University Medical Center, 2424 Erwin Road Suite 1102 Hock Plaza (Box 2721), Durham, NC 27710, USA; congwen.zhao@duke.edu (C.Z.); benjamin.a.goldstein@duke.edu (B.A.G.); 3Center for Predictive Medicine, Duke Clinical Research Institute, Duke University, 200 Morris Street, Durham, NC 27701, USA; arbrucke@ncsu.edu; 4Performance Services, Duke University Health System, 615 Douglas St Suite 600, Durham, NC 27705, USA; jen.massengill@duke.edu; 5Case Management, Duke University Health System, Dept 946, Durham, NC 27710, USA; patricia.kramer@duke.edu; 6Division of General Internal Medicine, Duke University, 2424 Erwin Road, Durham, NC 27705, USA; eric.poon@duke.edu

**Keywords:** hospitalization, patient readmission, clinical decision support systems, readmission risk model, risk assessment

## Abstract

Unplanned hospital readmissions represent a significant health care value problem with high costs and poor quality of care. A significant percentage of readmissions could be prevented if clinical inpatient teams were better able to predict which patients were at higher risk for readmission. Many of the current clinical decision support models that predict readmissions are not configured to integrate closely with the electronic health record or alert providers in real-time prior to discharge about a patient’s risk for readmission. We report on the implementation and monitoring of the Epic electronic health record—“Unplanned readmission model version 1”—over 2 years from 1/1/2018–12/31/2019. For patients discharged during this time, the predictive capability to discern high risk discharges was reflected in an AUC/C-statistic at our three hospitals of 0.716–0.760 for all patients and 0.676–0.695 for general medicine patients. The model had a positive predictive value ranging from 0.217–0.248 for all patients. We also present our methods in monitoring the model over time for trend changes, as well as common readmissions reduction strategies triggered by the score.

## 1. Introduction

Unplanned readmissions after hospitalization represents a significant value concern for American healthcare. Unplanned readmissions are associated with unnecessary costs and have been estimated to amount to USD 44 billion annually [1]. Readmissions are also associated with patient dissatisfaction [2], increased mortality, and increased length of stay [3]. Readmissions may indicate a poor quality discharge process. As an example of this, approximately half of patients that are readmitted have not seen a physician between their hospital discharge and their readmission [4]. Centers for Medicare and Medicaid Services (CMS) has tried to affect change in readmissions by developing financial disincentives to hospitals with higher-than-expected 30 day readmission rates. This is mandated through the Hospital Readmission Reduction Program [5], part of the Patient Protection and Affordable Care Act. These disincentives are financial penalties that negatively affect hospital revenues and can range up to 3% of a hospital’s total CMS inpatient payment. These penalties are also publically reported by CMS and can negatively affect a hospital’s public image [6]. 

Approximately 27% of readmissions are estimated to be preventable [7,8] and therefore much effort has been made to understand the factors that affect readmissions and how those factors could be used to predict which patients could be at risk for readmissions. There are a myriad of factors that can affect readmissions [9,10,11]. These include demographic and socioeconomic factors such as age, sex, race, insurance payer status, primary care provider availability, home location, marital or caregiver status, educational and healthcare literacy level. Clinical conditions clearly impact readmission risk and many of the chronic medical conditions all have independent associations with readmission risk. Clinical parameters such as physical/functional status, number and type of medications, vital signs, and laboratory values prior to discharge can affect readmissions. Prior health care utilization generally predicts readmissions as well.

At Duke University Health System (DUHS), early attempts to identify patients at high risk for readmission were based on a clinical decision support system (CDS) that incorporated many of the commonly understood variables that can affect readmission risk exported into a multivariable calculation to yield risk of readmission. However, this CDS system was only available after discharge so it was not helpful to the clinical teams to intervene with high risk patients prior to discharge. This has been a common problem for many readmission risk models as some of the data points for readmission models are administrative in nature and not available until after discharge [11,12]. We recognized the need for a system that (1) was part of our electronic health record (Epic systems) and (2) was able to produce a readmission risk score in a continuous real-time manner prior to discharge. 

While there has been a great rise in the usage of CDS tools, a prevailing challenge is how best to evaluate and continually monitor their performance [13]. In this article, we describe our efforts at DUHS in incorporating the Epic system readmission risk model into daily use on general medicine services. We describe our ongoing monitoring of the model’s performance to ensure continual acceptability. 

## 2. Materials and Methods

### 2.1. Study Environment

DUHS is located in North Carolina and consists of three hospitals: one tertiary (Duke University Hospital, DUH, Durham, NC, USA) and two community hospitals (Duke Regional Hospital, DRH, Durham, NC, USA and Duke Raleigh Hospital, DRAH, Raleigh, NC, USA), totaling 1500 inpatient beds. Since 2014, we have used a shared, Epic based electronic health record (EHR) system. Hospitalists performed 28,000 inpatient discharges at our 3 hospitals in 2019, with a 30-day unplanned readmissions rate for all adult patients of 10–12% across the three hospitals. 

Each hospital has their own case management structure, in which case managers coordinate discharge planning with clinical providers in either a team-based or unit-based configuration. Team-based configuration partners a case manager with a provider and the team has patients in many areas of the hospital. Unit-based configuration assigns a case manager to a specific unit in the hospital and the case manager may interact with several providers throughout the day as they round on patients on different units. Case managers are responsible for many aspects of discharge coordination, such as rehabilitation and home health referrals, community resource referrals, durable medical equipment orders, medication support, transportation, and other complex tasks. Case managers coordinate discussions around discharge needs with the patient (or their caregivers) and other clinical team members such as ordering providers (physicians or advanced care providers), nurses, physical or occupational therapists, and pharmacists. Case managers also depend on the Duke Resource Center to accomplish some of these tasks for discharge. The Duke Resource Center is an off-site phone call triage center which provides some basic functions of discharge and case management support.

### 2.2. Readmissions Risk Score

In response to needing better control of unplanned readmissions we sought to implement a CDS tool to predict risk of readmissions. After considering various options such as an internally developed machine learning model, an Epic based model and models from external vendors, we decided to implement the model native to our Epic system: “Unplanned readmission model version 1” [14]. Considerations included model quality, ease of implementation, ease of customization, and cost to operate and maintain. 

The readmission risk model uses immediately available data within the Epic EHR and calculates the risk of readmission on a linear scale from 0–100 every four hours during the hospital stay. The Epic proprietary model was derived from data from 4 hospitals and used LASSO penalized regression techniques to produce their final model. The final model variables includes patient age, clinical diagnoses variables, laboratory variables, medication numbers and classes, order types and utilization variables. Because of its proprietary nature, we are not permitted to list the model’s specific variables nor their weights. The model score is made available to hospital team members as an alert column that can be added to their patient lists they are working from. The score visually provides the risk score (0–100) and color codes the score red (high risk), yellow (medium risk), and green (low risk) based on the user-specified thresholds chosen for risk (Figure 1). This color coding scoring system allows users to quickly identify those patients at highest risk for readmission. The model has been reported to outperform one of the most established readmission prediction models, LACE [15], with an Area Under the Curve (AUC) for the Epic model = 0.69–0.74 compared to the LACE model = 0.63–0.69 [14].

### 2.3. Implementation Process

The Epic readmission risk model was implemented by DUHS in November 2017 and has been running continuously since then. Clinical providers and case managers on the general medicine service added the Epic readmission model score to their patient work lists along with other flags or alerts, which identifies tasks or issues for them to be aware of as they work. The expectation was that the clinical providers would discuss all high risk readmission score patients with the patient’s case manager to implement interventions prior to and after discharge to reduce the risk of readmission.

Based on discussions with case management and clinical providers, we estimated that interventions for preventing readmissions could be accomplished in 25% of all discharges. Therefore, we set the high risk threshold to identify the top 25% at risk for readmission. The next 25% would score medium risk, and the lowest 50% would score low risk. Therefore, we set the CDS based on capacity to provide the intervention, rather than metrics of sensitivity or positive predictive value. The Epic readmission risk model version that we used required fixed values as thresholds (rather than continuously updated quartiles). This necessitated ongoing threshold evaluation in case of a need to potentially reset thresholds as quartiles changed.

### 2.4. Evaluation

We formed a quarterly working group consisting of representatives from Hospital Medicine, Case Management, Clinical Operations, Biostatistics and Informatics. This working group reviewed the performance of the risk score and adoption of the CDS. The primary outcome that we evaluated the risk score against was its ability to predict unplanned readmissions. We defined an unplanned readmission and index admission as follows:Both index hospitalization and readmission are to any DUHS facility. Readmissions occurred within 30 days of the index hospitalization’s discharge date.Index hospitalization was between dates 1 January 2018–31 December 2019 and included all patients that were inpatient status, and ages 18 and older, admitted to any of the three DUHS hospitals. We excluded patients whose index admissions were based on psychiatric diagnoses, rehabilitation care, non-surgical cancer MSDRGs (Medicare Severity Diagnosis Related Groups), or admitted for inpatient hospice. Patients who were transferred to other acute facilities, died during index hospitalization, or left against medical advice were also excluded.Readmissions are hospitalization within 30 days of discharge from an index hospitalization and included patients age 18 and older with inpatient status. We excluded patients whose readmission was based on psychiatric diagnoses, rehabilitation care, or who had a planned readmissions (based on the CMS algorithm) [16].

We designed a standard template to report cumulative patterns in readmissions, score distribution, and interventions as well as the statistical performance of the model. This regular monitoring was needed to ensure the thresholds were maintaining the desired 25–25–50% breakdown of high-medium-low risk groups and adjust them if the score distribution shifted significantly. The report also included measures of the score’s statistical discrimination and calibration within and across hospital and service line cohorts, to assess whether its performance was improving over time as the score became more tailored to the DUHS population. Additionally, data on interventions allowed the team to understand and plan strategies in reducing readmissions. 

In this paper, we focus on the performance of the risk score from 1 January 2018–31 December 2019. Since a patient will have multiple risk scores over the course of an admission—one every 4 h—we decided to evaluate the score based on a patient’s maximum score over the course of the admission. We assess the performance of the overall risk score based on AUC and calibration slope [17], the stability of the decision thresholds over time, and the performance of the decision rules based on sensitivity, specificity, and positive predictive value. We evaluate the performance of the risk score and associated CDS across each hospital, as well as within the general medicine, heart and oncology service lines. 

### 2.5. Institutional Review Statement

This study was approved by the Institutional Review Board (IRB) of Duke University as exempt without need for informed consent. 

## 3. Results

For the two year period between 1 January 2018 and 31 December 2019, 112,409 adult patients were discharged from the three DUHS hospitals (Table 1) Across all adult patient discharges, the unplanned readmission rate (within the health system) ranged from 10–12%. The unplanned readmission rate during this time frame was stable for all three hospitals (Figure 2).

The median readmission model score ranged from 13–14. Using data from the first quarter of 2018, we defined low-medium and medium-high thresholds by choosing values that would identify the top 25% of risk scores as high risk, and the next 25% as medium risk and the lowest 50% as low risk. The low to medium risk threshold was 14 and medium to high risk threshold score was 21. These risk thresholds remained stable over the time period (Figure 3). 

Table 2 shows the performance metrics for the readmission model. Overall, the model’s discriminatory performance was good with an AUC of 0.716–0.760 for all adult patients at the three hospitals. Calibration of the model was good with a calibration slope close to 1.0 (Table 1). Service line differences reveal lesser discriminatory performance for general medicine, but the AUC was still in the acceptable range of 0.676–0.695 (Table 2). The positive predictive value of a high risk score ranged from 0.0217 to 0.248 for all patients, which is significantly higher than baseline readmission rates, indicating the usefulness of the model in identifying which patients to focus readmission interventions on. Figure 4 shows the variability of AUC over the two year study period. The model AUC performance was relatively stable, with less than 7% variability.

We also tracked interventions performed by the clinical team (mainly case managers and providers) prior to discharge for the general medicine service. This intervention data were not available for other service lines. These interventions are described in 4. Discussion (below) and shown in Table 3. As expected, patients who received any intervention to prevent readmission had a higher risk score than those who had no interventions (Figure 5).

## 4. Discussion

CDS tools have become a routine and important component of clinical care provision. While significant work often goes into developing CDS tools, it is important to monitor a tool’s performance prospectively. Hospital readmissions is one area that has seen applications of CDS. It is a complex issue for both inpatient and outpatient systems to address, and it is difficult to identify accurately which patients are at highest risk to focus interventions on. Interventions themselves can be costly and time-consuming and it is not feasible to apply intensive readmission prevention strategies to all patients being discharged.

Current readmission models that have been published and validated utilize data mostly available at discharge and data from administrative data sets. These risk models are typically not available to the inpatient teams prior to discharge and therefore patients at highest risk for readmission would not be identifiable prior to discharge. In order to intervene on patients at high risk for readmission it is important to identify those patients prior to discharge [12,18]. Unfortunately, prediction models for readmission often perform worse prior to discharge as many of the highly predictive variables needed to predict readmissions (such as hospital length of stay) are available only at or after discharge [12,18]. The ideal readmissions scoring system for hospital based teams is one that is closely integrated with the electronic health record; pulls data real-time from the EHR, updates continuously, and presents the score clearly for the clinical teams to act upon. Prediction models that run outside of the EHR and require exporting of data to an external model for calculation and then importing the results add a layer of complexity that can impede adoption of the model.

In this study, we have shown that the Epic unplanned readmission model performs well on general medicine patients, oncology patients, and cardiology patients at three very different hospitals within our health system. The area under the curve (AUC or C-statistic) for overall adult patients of 0.716–0.760 and for general medicine of 0.0676–0.695 is very similar to published data for AUC for published readmission models for medicine or combined medicine/surgery populations (14 studies with published AUCs ranging 0.60–0.836 with an average of 0.726) [9,12,15,18,19,20,21,22,23,24,25,26,27,28]. We found the Epic model performed well for our populations of general medicine, cardiology, and oncology, and less well for surgical services. Application of the Epic model allowed the predictive probability of readmission to increase from 14–17% (overall average of general medicine readmissions at three hospitals) to 21.8–27.4%. This is a modest changed in predictive value when applied to our general medicine daily census of 400 patients at the three hospitals, and allows us to identify which patients in that large population to focus on. Epic has produced validated data experience with the model but we are unaware of other institutions publishing their experience over years of use of the model as we have.

We created an ongoing monitoring and maintenance program for the performance of this readmission model over the 2 years that we have been using it. That process included quarterly template reports that help us look at the daily score trends, review predictive statistics of the model, and make adjustments to the thresholds identifying low, medium, or high risk patients. The team that was performing this review was multidisciplinary, involving physicians, case managers, statisticians, and project managers. We undertook this review process as part of ensuring the model was performing as expected. We believe our work is novel in that we can find no other published efforts reporting AUC performance over time for similar risk models. In this article, we report on the AUC performance being relatively stable over the two year period of the study (Figure 4). 

The readmissions scoring model has allowed our general medicine clinical teams to arrange the following interventions for patients at the highest risk for readmission prior to or at discharge (Table 3):Case management discussion with clinical providers to refer patient to intensive case management referral services after discharge.Clinical team obtaining a hospital follow-up visit scheduled within 7 days of discharge.Pharmacist collaboration with discharging team to perform medication reconciliation prior to discharge.Duke Resource Center calls patient within 48 h of discharge and performs a post-discharge phone call.

Our results should be interpreted with an appreciation of some of the limitations inherent in this work. As far as applicability to other institutions, the model is likely to perform differently and there should be attempts to understand the institution-specific performance of the model as it is considered for implementation. We were unable to adjust the model to emphasize variables that may be more specific to DUHS. For example, one of the variables is “any prior order” for EKG, which is probably not contributing to predictability much, as all admitted patients at DUHS will likely have an order for EKG. Our understanding is that the “customization” of the Epic readmission model variables is possible, and we have not pursued that yet. Additionally, we are unable to adjust risk model thresholds for different service lines or hospitals, so we have only the one model covering different patient populations with different performance specifics. We also only looked at readmissions that occurred at our three affiliated hospitals (e.g., “same-system readmissions”) and therefore missed the ability to capture readmissions at other local hospitals. That data capture would have required payer claims data which we did not have available. In addition, we did not review length of stay or other quality or financial measures that could have been negatively impacted by this study. However, despite these limitations, we remain satisfied with the model because: 1. It has reasonable discriminatory performance that is maintained over time; 2. It allows real-time risk assessment of inpatients’ risk of readmission prior to discharge; 3. It is easily adopted into clinical workflows given its integration in the Epic electronic record. The Epic readmission risk model has allowed our general medicine clinical teams to identify patients at DUHS most at risk for readmission. This has allowed us to collaboratively work with our multidisciplinary teams to bring forward interventions designed to reduce the risk of readmission and improve the value of care delivered.

## Figures and Tables

**Figure 1 jpm-10-00103-f001:**
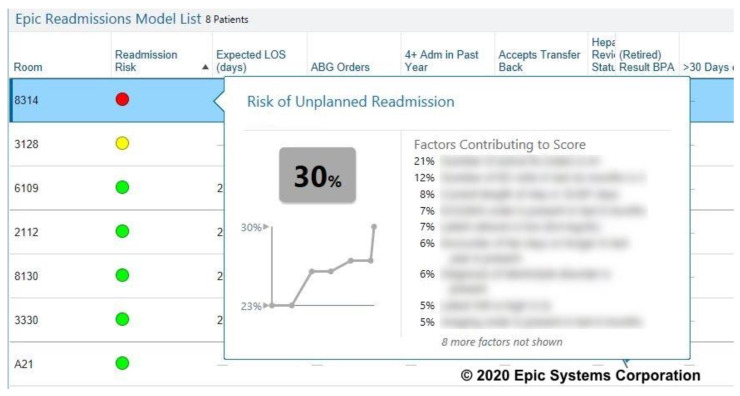
Epic Readmission Risk Model Appearance in Patient Lists. Risk variables blurred at request of Epic.

**Figure 2 jpm-10-00103-f002:**
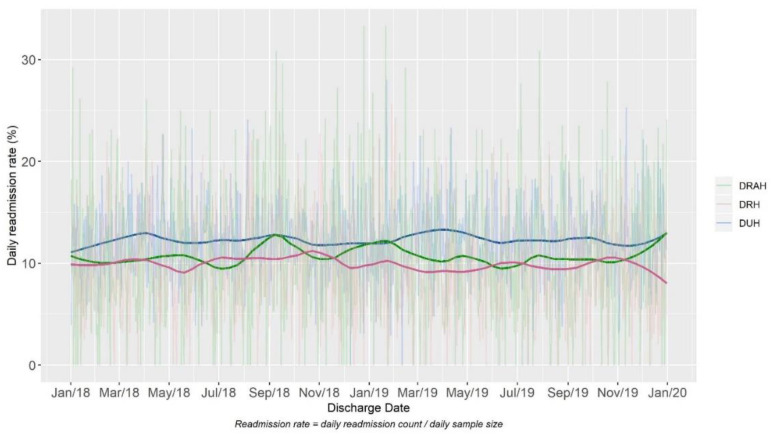
Readmission rate per day from 1 January 2018–31 December 2019 at three DUHS hospitals. Plots show a LOWESS smoothed curve in bold of the trend over time with the unsmoothed daily readmission rates behind.

**Figure 3 jpm-10-00103-f003:**
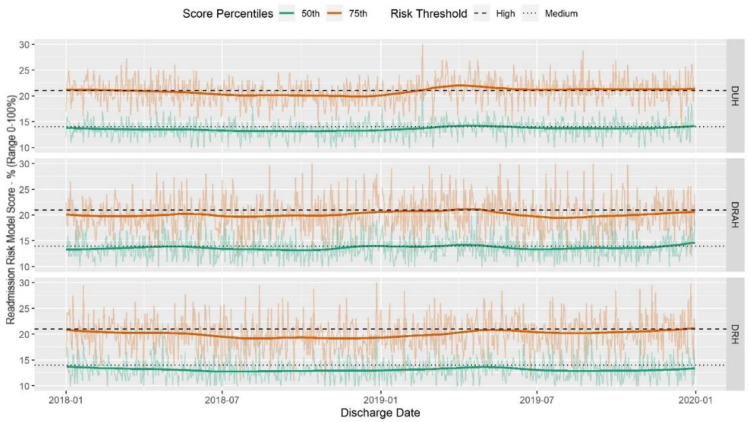
Daily encounter-level risk scores from 1 January 2018–31 December 2019 at three DUHS hospitals. Plots show a LOWESS smoothed curve of the trend over time in bold with the unsmoothed daily readmission rates behind.

**Figure 4 jpm-10-00103-f004:**
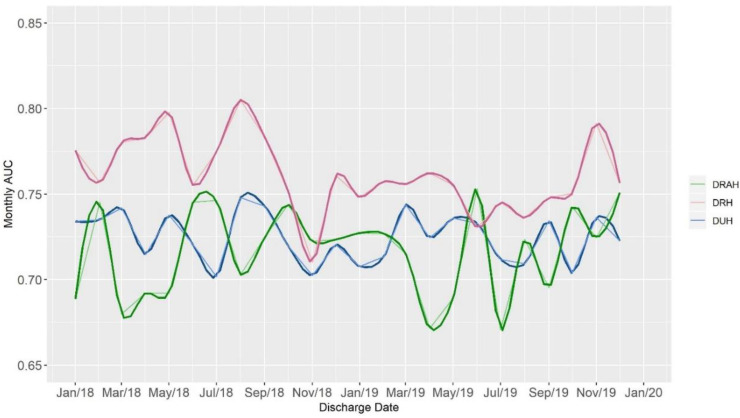
Area under the curve (AUC) over time by hospital. Figure shows LOWESS smooth in bold over time with the unsmoothed data below.

**Figure 5 jpm-10-00103-f005:**
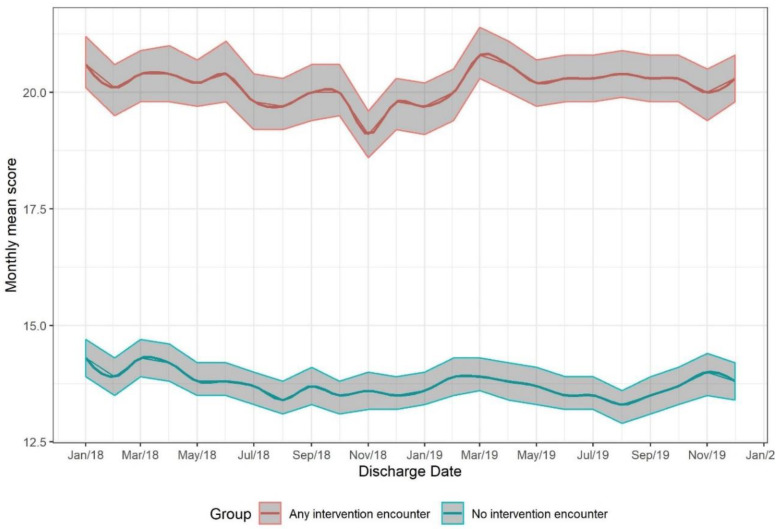
Mean risk score (with 95% CI) over time by whether intervention was received or not.

**Table 1 jpm-10-00103-t001:** Characteristics in Encounter-Level from 01/01/2018–12/31/2019.

	DUH	DRH	DRAH
Number of discharges	67,219	27,405	17,785
Median score (IQR)	14 (9–21)	13 (7–20)	13 (9–20)
Readmission number (rate)	8308 (12%)	2729 (10%)	1905 (11%)
Number (percentage) of patients getting any intervention	31,552 (47%)	12,671 (46%)	4698 (26%)
AUC	0.72	0.72	0.76
Calibration Slope	1.03	1.07	0.97

DUH = Duke University Hospital, DRH = Duke Regional Hospital, DRAH = Duke Raleigh Hospital, AUC = Area Under Curve, IQR = Interquartile Range.

**Table 2 jpm-10-00103-t002:** Performance Metrics for Readmission.

Population	AUC	Readmission Rate	Positive Rate at Medium Risk ^1^	Positive Rate at High Risk ^2^	PPV at Medium Risk	PPV at High Risk	Negative Rate at Medium Risk	Negative Rate at High Risk
**DUH Overall**	0.725	12%	0.249	0.534	0.128	0.248	0.239	0.228
**DUH Gen Medicine**	0.694	17%	0.229	0.631	0.130	0.274	0.315	0.345
**DUH Heart**	0.707	13%	0.263	0.599	0.108	0.232	0.326	0.297
**DUH Oncology**	0.611	22%	0.272	0.637	0.194	0.262	0.312	0.496
**DUH Surgery**	0.663	11%	0.297	0.299	0.139	0.224	0.230	0.129
**DRAH Overall**	0.716	11%	0.273	0.507	0.118	0.217	0.246	0.219
**DRAH Gen Medicine**	0.676	14%	0.283	0.580	0.126	0.218	0.320	0.339
**DRAH Heart**	0.680	8%	0.395	0.184	0.114	0.179	0.267	0.073
**DRAH Oncology**	0.513	20%	0.429	0.429	0.250	0.200	0.321	0.429
**DRAH Surgery**	0.688	11%	0.219	0.369	0.117	0.311	0.202	0.100
**DRH Overall**	0.760	10%	0.243	0.567	0.109	0.227	0.219	0.214
**DRH Gen Medicine**	0.695	14%	0.245	0.636	0.111	0.231	0.328	0.351
**DRH Heart**	0.613	9%	0.387	0.484	0.073	0.150	0.477	0.265
**DRH Oncology ^3^**	NA	NA	NA	NA	NA	NA	NA	NA
**DRH Surgery**	0.737	10%	0.268	0.366	0.144	0.261	0.170	0.110

^1^ Medium risk: risk score >= 14 and risk score < 21, ^2^ High risk: risk score >= 21, ^3^ DRH oncology not separately available; DUH = Duke University Hospital, DRH = Duke Regional Hospital, DRAH = Duke Raleigh Hospital, AUC = Area Under Curve, PPV = Positive Predictive Value.

**Table 3 jpm-10-00103-t003:** Average readmission risk scores (percentage of encounters) for interventions.

Intervention Type	DUH	DUH General Med	DRAH	DRH
No Intervention	14.09 (53%)	19.27 (34%)	15.04 (74%)	11.82 (54%)
Any Intervention	19.83 (47%)	22.52 (66%)	20.36 (26%)	20.89 (46%)
Arranged transportation	20.74 (26%)	22.4 (48%)	21.73 (12%)	21.08 (36%)
Arranged HH visits	21.64 (13%)	23.64 (17%)	18.47 (3%)	21.81 (15%)
Referral to SNF	22.51 (10%)	23.95 (19%)	20.78 (16%)	23.87 (11%)
Procured DME	17.06 (11%)	21.53 (7%)	14.78 (1%)	17.51 (7%)
Medication assistance/support	17.49 (2%)	18.81 (3%)	19.39 (1%)	17.65 (2%)
Family training for elder patients	23.6 (1%)	23.51 (3%)	24.52 (0%)	23.93 (3%)
Geriatrics follow-up	22.18 (1%)	22.93 (4%)	0 (0%)	0 (0%)
Arranged outpatient dialysis	30.97 (1%)	33.57 (2%)	27.12 (0%)	33.23 (0%)
Duke Well (outpatient CM)	27.53 (0%)	28.32 (1%)	35 (0%)	24.24 (1%)

HH = Home health, SNF = Skilled nursing facility, DME = Durable medical equipment, CM = Case management, DUH = Duke University Hospital, DRH = Duke Regional Hospital, DRAH = Duke Raleigh Hospital.

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
