# Peer review of "Implementation and Continuous Monitoring of an Electronic Health Record Embedded Readmissions Clinical Decision Support Tool"

_jpm, 2020, doi:10.3390/jpm10030103_

Round 1

Reviewer 1 Report

Reviewing the manuscript entitled, “Implementation and continuous monitoring of an electronic health record-based readmissions scoring model.” By Gallagher D. et al., this is a well-written article about implementation of their developed electronic health record-based readmissions scoring model. The new HER-based readmissions scoring system that they proposed overcomes the shortcomings of current models, reveals that their AUC is around 0.7, which indicates that this has a high judgment ability. This model will be a very effective tool in the medical economy, especially in a super-aging society. So, when the authors respond to my concerns, this will reach to an acceptable quality.

Concerns

At line 280 in the discussion, as you have mentioned “It allows real-time risk assessment of inpatients’ risk of readmission prior to 280 discharge,” I completely agree with it. This is the major point of your model. On the other hand, as you have mentioned from line 275 to line 278, it is extremely important to cooperate with local hospitals for the development of this system. For further growth, the author should describe the ability to capture readmissions at other local hospitals in detail.

Figure 2 and 3 are difficult to understand. The authors should modify Figure 2 and 3, and their legends.

Reviewer 2 Report

This is a very interesting study and certainly well thought out. The authors present important information. I am sure that some critics may suggest that the something is amiss with the study. However, that is what research should be about.

Author Response

No edits or changes requested by reviewer

Reviewer 3 Report

The authors of this manuscript present the results of the implementation and monitoring of an Electronic Health Record-Based Readmissions over a two years period in three hospitals. They also analyse their methods in monitoring, as well as the readmissions reduction strategies triggered by the score.

The topic is of great interest to the medical community, as it proves the applicability of an internally developed machine learning model to improve healthcare given to hospitalized population before discharge. However, several issues need to be addressed to clarify this manuscript.

Major issues:

  1. Page 2, line 76. Materials and Methods, Study Environment. The authors describe the management structure in either a team-based or unit-based configuration. Would be helpful if the authors give some extra information of those team or unit-based configuration (table or supplementary material).
  2. Page 2, line 94-99. Materials and Methods. Readmissions Risk Score. The authors list the final model variables, previously they mention the use of ‘clinical decision support system (CDS) that incorporated many of the commonly understood variables that can affect readmission risk’. I would suggest the authors adding more detailed information, even in a table or supplementary material.
  3. Page 3, Figure 1. Factors Contributing to Score. Only percentages can be read, text is blurred. This information is of great interest for the readers. I understand that those variables are part of the final model defined previously at Readmissions Risk Score. Abbreviations are not defined at footnote.
  4. page 4. Evaluation. The authors define unplanned readmissions and index admissions, but I see well defined the exclusion criteria. I would suggest the authors a more detailed description of the inclusion criteria. Besides that, would be of great interest to know all Medical Specialties’ included.
  5. Page 4, line 162. Institutional Review Statement. ‘…as exempt’. Does it mean informed consent was not necessary? please clarify.
  6. Page 5, lines 176-180. The authors describe data from Table 1, but Median Score (IQR) is not clearly explained at Material and Methods.
  7. Page 7, line 210. At Table 3, authors describe DUH General Medicine. It would be good to know why the authors did not describe the data of Intervention Type of other centers and other populations. AUC was higher in Heart (DUH) or Surgery (DRH). Readmissions were high in Oncology (DUH and DRAH).
  8. Page 9, line 257. Authors describe well the interventions for patients at highest risk for readmission in the setting of general medicine. It would be of great interest the authors describe what interventions were arranged at the other specialties as oncology and cardiology.
  9. Page 10, line 279. After the limitations, the authors conclude that they remain satisfied. I would suggest the authors should remark how their work supported healthcare coordination, improving the quality of the care system.
  10. Did the authors measure the impact on length of stay? as the interventions started during admission could have delayed the discharge. Also, of interest is the economic impact of those interventions, did the authors analyze this data?

Minor issues:

  1. Page 2, line 46. ‘PCP’, is not previously defined.
  2. Page 2, line 72, ‘electronic health record (EHR)’, on lines 59 and 92 I suggest the authors using the same abbreviation.
  3. Page 2. Line 52, ‘Duke University Health System (DUHS)’, after, at lines 63 and 69 the authors did not use the abbreviation given before.
  4. Page 2, line 80. ‘Duke Resource center’, Center with capital letter.
  5. Page 4, line 136. ‘cancer MSDRGs’, the abbreviation is not previously defined
  6. Page 4, line 142. CMS algorithm. I would suggest the authors describe this algorithm or refer it.
  7. Page 4, line 170. Table 1. I would suggest the authors to define abbreviations under the table (DUH, DRH, DRAH, IQR, AUC).
  8. Page 5, Figure 3. Missing data on left side number and units.
  9. Page 7, line 197. Table 2. Authors should define abbreviations used at the table.
  10. Page 8, lines 209-210. Table 3. Are the numbers the risk score? Please clarify units.

        Authors should define abbreviations used at the table

  1. Page 9, line 264. Duke Resource center, Center with capital letter.
  2. Page 9, line 270. ‘Duke University Health System’, the authors should use the abbreviation DUHS.

Round 2

Reviewer 3 Report

The authors have put a considerable amount of effort in revising this manuscript, according to the reviewer's instructions. They have made the clarifications as requested improving the content of this interesting work.